# Dispersal patterns of *Trypanosoma cruzi* in Arequipa, Peru

**Alexander S. F. Berry**[1,2], **Renzo Salazar-Sánchez**[3], **Ricardo Castillo-Neyra**[3,4],
**Katty Borrini-Mayorí**[3], **Claudia Arevalo-Nieto**[3], **Claudia Chipana-Ramos**[3], **Melina Vargas-Maquera**[3], **Jenny Ancca-Juarez**[3], **César Náquira-Velarde**[3†], **Michael Z. Levy**[3,4],
**Dustin Brisson**[2]*

1 Division of Gastroenterology, Hepatology, and Nutrition, Children's Hospital of Philadelphia, Philadelphia, Pennsylvania, United States of America, 2 Department of Biology, University of Pennsylvania, Philadelphia, Pennsylvania, United States of America, 3 Zoonotic Disease Research Lab, One Health Unit, School of Public Health and Administration, Universidad Peruana Cayetano Heredia, Arequipa, Peru, 4 Department of Biostatistics, Epidemiology and Informatics, The Perelman School of Medicine of the University of Pennsylvania, Philadelphia, Pennsylvania, United States of America

† Deceased.
* dbrisson@sas.upenn.edu

**Data Availability Statement:** All reads are available at Sequence Read Archive (SRA) under the BioProject PRJNA517719.

## Abstract

Anthropogenic environmental alterations such as urbanization can threaten native populations as well as create novel environments that allow human pests and pathogens to thrive. As the number and size of urban environments increase globally, it is more important than ever to understand the dispersal dynamics of hosts, vectors and pathogens of zoonotic disease systems. For example, a protozoan parasite and the causative agent of Chagas disease in humans, *Trypanosoma cruzi*, recently colonized and spread through the city of Arequipa, Peru. We used population genomic and phylogenomic tools to analyze whole genomes of 123 *T. cruzi* isolates derived from vectors and non-human mammals throughout Arequipa to determine patterns of *T. cruzi* dispersal. The data show significant population genetic structure within city blocks—parasites in the same block tend to be very closely related—but no population structure among blocks within districts—parasites in neighboring blocks are no more closely related to one another than to parasites in distant districts. These data suggest that *T. cruzi* dispersal within a block occurs regularly and that occasional long-range dispersal events allow the establishment of new *T. cruzi* populations in distant blocks. Movement of domestic animals may be the primary mechanism of inter-block and inter-district *T. cruzi* dispersal.

## Author summary

Urbanization creates environments that are ideal for some human pests and pathogens. As the number and size of urban environments increases globally, it is becoming vital to understand how human disease-causing pathogens, their vectors, and their non-human hosts disperse through urban landscapes. Here we study a population of *Trypanosoma cruzi*–the protozoan parasite and causative agent of Chagas disease in humans–that

**Funding:** This work was funded by grants from the National Institute of Health (https://www.nih.gov/) (T32HG000046 (AB), T32AI007532 (AB), R01AI142572 (DB), R01AI097137 (DB), and NIAID 5R01AI101229 (MZL)), the National Science Foundation (https://nsf.gov/) (DEB-1354184 (DB)), and Burroughs Wellcome Fund (https://www.bwfund.org/) (1012376 (DB)). The funders had no role in study design, data collection and analysis, decision to publish, or preparation of the manuscript.

**Competing interests:** The authors have declared that no competing interests exist.

recently colonized the city of Arequipa, Peru. We use population genomic and phylogenomic tools to understand how this parasite population dispersed through the city to achieve its current distribution and abundance. We show that *T. cruzi* collected from the same city block tend to be very closely related, while those from neighboring blocks are often as distantly related as those from blocks in distant districts. The data suggest that vectors facilitate frequent within-block dispersal of the parasite, while domestic animal movement may facilitate the relatively infrequent inter-block and inter-district dispersal.

## Introduction

Human populations are increasingly moving from rural to urban centers resulting in fundamental habitat alterations that have caused local extinctions of many native species. This influx of humans and domestic animals also creates suitable habitats that can be colonized by invasive species, including human pests and pathogens. Given the current rate of immigration and urbanization, the geographic distributions of many pathogens will depend on the manner in which their hosts and vectors interact with the novel biotic and abiotic features of growing cities [1].

Proliferation in an ecosystem is accomplished by immigration to a new location, establishment (reproduction and population growth), and dispersal to new areas within the ecosystem [1]. Investigations into the dynamic processes of immigration, establishment, and dispersal of disease-causing agents in human-altered ecosystems are particularly important for public health risk management because the distribution and abundance of these pest species is correlated with the incidence of disease in humans [2]. In this study, we examine the dispersal pattern of *Trypanosoma cruzi*, the causative agent of Chagas disease, around the region of Arequipa, Peru.

The Chagas disease system in Arequipa, Peru is ideal to study pathogen dispersal in a rapidly changing environment for several reasons. First, a single introduction of *T. cruzi* recently established in Arequipa and has since dispersed to many districts in the city [3]. A single population allows dispersal patterns through the city to be assessed without being confounded by repeated immigration from other geographic regions. Second, the expansion of urban and semi-urban environments over the past ~50 years [4,5] has resulted in a mosaic of several habitat types, including different quantities of domestic animals and different husbandry practices, each of which provides unique challenges and opportunities for *T. cruzi* to establish and proliferate [5–7]. Lastly, this system is well studied, permitting hypotheses about the proximal causes of dispersal patterns to be formed. For example, dispersal of *Triatoma infestans*, the only vector of *T. cruzi* in Arequipa, is hindered by roads between city blocks, making it unlikely that the vector facilitates *T. cruzi* block-to-block migration [6]. Additionally, the number of domestic animals in and around each house in Arequipa is known, which will allow us to correlate the migration patterns of *T. cruzi* with the presence and abundance of important reservoir hosts in future studies. Here, we investigate how a zoonotic pathogen has dispersed across this complex urban landscape by analyzing the genome sequences of 123 *T. cruzi* isolates collected throughout the city of Arequipa, Peru.

## Results

### *T. cruzi* cluster within city blocks, not within districts

123 DTU 1 *T. cruzi* parasites, representing a single, closely related population [3], were collected from ten geographically-distinct districts throughout the region of Arequipa, Peru (Fig

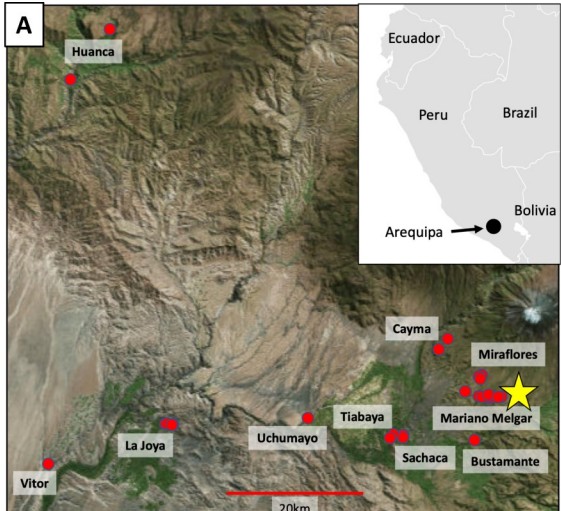

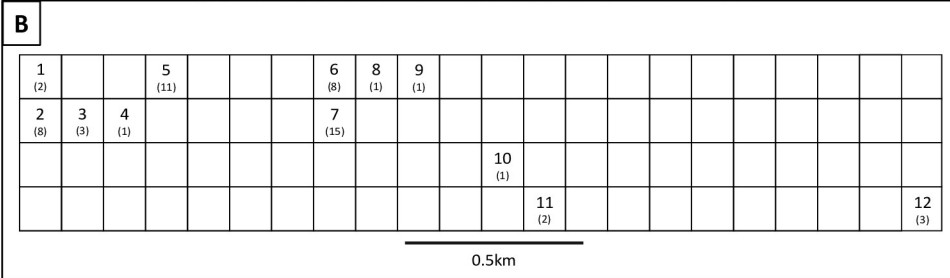

**Fig 1. Map of sample collection locations in A) Arequipa, Peru and the B) Mariano Melgar district. A)** All 123 *T. cruzi* samples used in this analysis were collected within 100km of Arequipa, Peru from 2008–2015. Sampled *T. cruzi* isolates are represented by red dots. Mariano Melgar—the city center and most densely-sampled district—is indicated with a yellow star. Inset shows the location of Arequipa in southern Peru. Map of Arequipa was obtained from https://landlook.usgs.gov, sample locations were plotted using QGIS v. 3.4 [8], and inset was modified from https://commons.wikimedia.org/wiki/Atlas_of_the_world [9]. **B)** Blocks from which *T. cruzi* were collected in the Mariano Melgar district (N = 56) are numbered 1–12 with the number of samples collected per block in parentheses. No parasites were collected in unlabeled blocks. Blocks are displayed in a grid to maintain privacy with black lines representing streets separating blocks.

1). Multiple isolates were sequenced from 20 of the 37 blocks in Arequipa where *T. cruzi* were found. On only three of these blocks were all of the isolates sequenced cultured from different bugs collected from a single house, although the isolates were collected in different years in two of these cases. On 17 blocks, an average of 5 isolates were sequenced which were cultured from bugs collected from an average of 3 houses per block. The common ancestor of the Arequipan *T. cruzi* population was estimated to live as recently as 8.8kya, however there is significant uncertainty in estimates of divergence times [3]. A total of 9271 of the >28 million sites in the genome sequences were polymorphic from the 123 *T. cruzi* parasites. The average pairwise genetic distance between all samples was 1726 SNPs (Table 1). The average pairwise distance between samples from different houses within city blocks, however, was considerably smaller (620 SNPs). Further, the average pairwise distance between samples from different blocks within each district was much greater (1925 SNPs) than within-block genetic differences (Fig

**Table 1. Population Genetic Statistics.**

| | Avg. Pairwise distance | No. Segregating Sites | π | θ |
|---|---|---|---|---|
| 123 *T. cruzi* isolates | 1761.96 | 12256 | $3.45*10^{-5}$ | $4.46*10^{-5}$ |

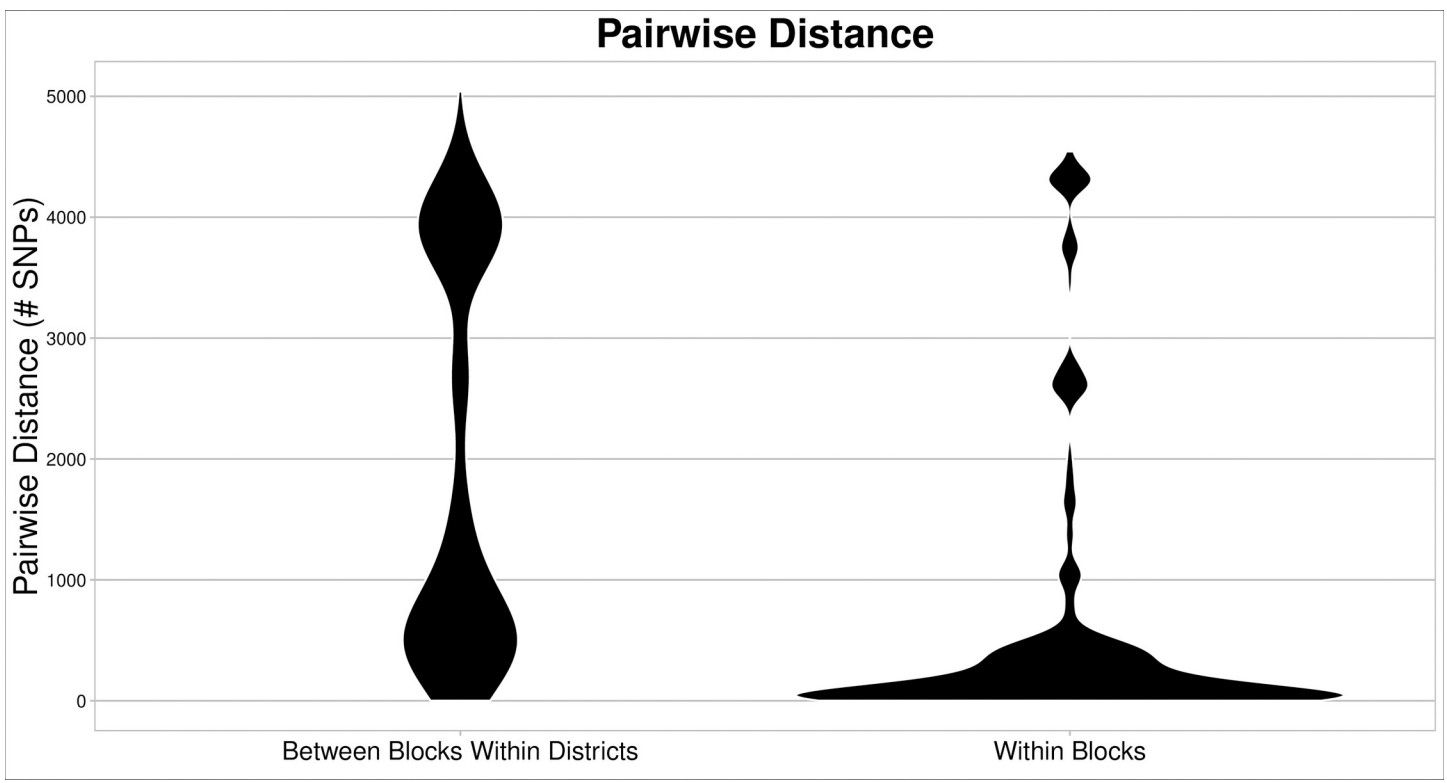

**Fig 2. *T. cruzi* isolates collected from the same block are genetically similar.** Violin plots represent the number of pairwise SNP differences among pairs of samples collected in different blocks from the same district (left) and among pairs of samples collected in the same block (right). The width of the violins represents the frequency that pairwise distances were detected. Samples found in different blocks within a district are significantly more different from one another than samples collected within the same city block (p<0.001). Genetically-similar samples tend to cluster within blocks despite each district containing much of the overall genetic diversity among all samples.

2). There is significantly less variation among samples from different houses within blocks than expected (p<0.001) given the total diversity among samples (Fig 2; Table 2). The limited genetic diversity within each block observed in population genetic analyses is also evident in phylogenetic analyses as most city blocks contain a single *T. cruzi* lineage where all within-block diversity was generated by mutations after colonization of the block (Fig 3, S1 Appendix). This result is also supported by ADMIXTURE analyses of the 56 samples from the densely-sampled Mariano Melgar district, confirming that most city blocks are dominated by a single *T. cruzi* genetic cluster representing a single lineage (S1 Fig). Additionally, although sampling is limited, we find no evidence of temporal variation.

*T. cruzi* collected from the same block tend to be very closely related compared to parasites found in different blocks (Fig 2). *T. cruzi* gene flow is significantly restricted among blocks in

**Table 2. There is significant population genetic structure among blocks within districts, but not among districts.**

|  | *P-value* | *% total variation* |
|---|---|---|
| *Variation within blocks* | 0.001 | 27.828949 |
| *Variation among blocks within districts* | 0.001 | 67.370526 |
| *Variation between districts* | 0.41 | 4.800524 |
| *Variation within blocks in Mariano Melgar* | N/A | 27.44506 |
| *Variation between blocks in Mariano Melgar* | 0.001 | 72.55494 |

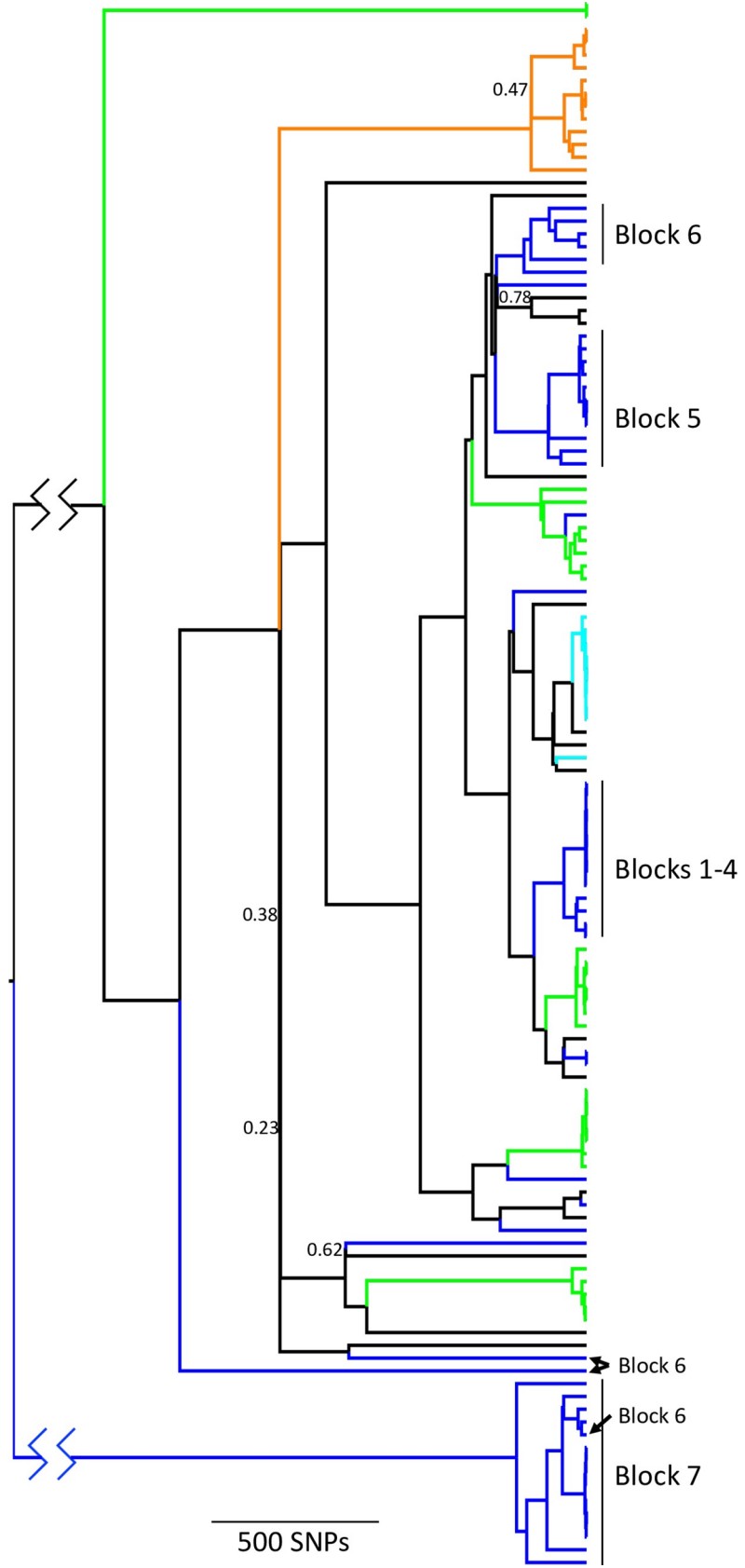

**Fig 3. *T. cruzi* collected from the same block often forms a monophyletic group.** The maximum clade credibility phylogeny includes all 123 *T. cruzi* samples collected from Arequipa with samples from Mariano Melgar labeled blue and those from La Joya labeled green. Block numbers are labeled for the 48 samples collected from Blocks 1–7 in Mariano Melgar. Black branches represent samples collected from the other districts in the Arequipa region: Bustamante, Cayma, Huanca, Miraflores, Sachaca, Tiabaya, Uchumayo, and Vitor. *T. cruzi* from each district, for example, Mariano Melgar and La Joya, span the phylogenetic diversity in the region suggesting that most of the genetic variation in Arequipa can be found within each single district. The *T. cruzi* in many blocks within Mariano Melgar were likely introduced from another district as their closest relatives were detected in other districts and not in other blocks within Mariano Melgar. Jagged lines in most basal branch indicate that the branch was shortened for visualization. Posterior probabilities less than 0.80 are labeled for basal nodes. A modifiable phylogeny containing all tip labels and posterior probabilities can be found in S1 Appendix.

analyses of the whole dataset and in analyses focusing only on Mariano Melgar ($p < 0.001$; Table 2). This is consistent with the result that *T. cruzi* in different blocks within Mariano Melgar belong to unique genetic clusters (S1 Fig, S2 Fig). Further, *T. cruzi* lineages occupying neighboring city blocks are often phylogenetically divergent such that geographic distance is not correlated with evolutionary distance (Fig 3).

There is no evidence supporting restricted gene flow among the districts of Arequipa (Table 1). The majority of the total genomic diversity is present in the two most densely-sampled districts—Mariano Melgar and La Joya (Fig 3). Further, the lineages occupying city blocks within Mariano Melgar or La Joya are commonly most closely related to lineages in other districts, suggesting dispersal among districts in the region is not sufficiently restricted to result in population genetic structure.

## Genetic linkage among SNPs

Many of the genomic differences among strains are the result of independent assortment of chromosomes or cross-over events that occur during sexual reproduction [9,10]. To account for the potential lack of independence among SNPs in the analyses, a dataset containing the presence or absence of 474 unique recombination events was generated as previously described [9]. All analyses of this dataset showed that using presence/absence of recombination events resulted in nearly identical estimations of relatedness and support for conclusions as the SNP dataset, despite the loss of power. The similarities of the results among these datasets likely results from the identity-by-decent of the recombination events (S1 Table; S3 Fig), making these events evolutionarily informative markers.

## Animal husbandry and migration

The number of guinea pigs imported from markets and farms was evaluated for 383 households across a transect in Mariano Melgar [11]. Interestingly a large number of guinea pigs were imported from a market into the blocks in which we detected multiple independent *T. cruzi* introductions. The distribution of guinea pigs is also correlated with the density of *Triatoma infestans* vectors, as well as the presence of *T. cruzi* in vectors. No correlation between the vector nor parasite population and either dogs or chickens was identified (S4 Fig).

## Discussion

Investigations into the impact of environmental change have focused primarily on habitat destruction and extinction of natural fauna [12–17]. Equally impactful is the creation of novel habitats that can be exploited by non-native species including pests associated with human-altered habitats [2]. Understanding how native and non-native species exploit human altered habitats, especially in human-dense urban habitats, is critical to understanding the distribution and abundance of human diseases. Here, we investigate the dispersal patterns of *Trypanosoma*

*cruzi*, the cause of human Chagas disease, through the urban ecosystem of Arequipa, Peru. The genomic data support the previously described pattern of migration in which dispersal proceeds with little impediment among houses within a city block soon after colonization of the block (Figs 2 and S1) [5]. For example, the *T. cruzi* population on Block 5 was founded by a single lineage in 2010 and could be collected from multiple houses throughout the block by 2011 (Fig 3) [5]. Establishing a novel *T. cruzi* population in a city block, however, is a rare event as no *T. cruzi* were detected in a majority of blocks and blocks harboring a population tend to contain only a single genetic lineage suggestive of a single colonization event (Fig 3). Interestingly, geographic proximity to blocks harboring a *T. cruzi* population increases the establishment probability of that block only slightly (Figs 3 and S1). Further, the *T. cruzi* lineages in eight of the 12 blocks investigated in the Mariano Melgar district were most closely related to *T. cruzi* found only in a geographically distant district of the region suggesting that inter-district immigration is not less likely than immigration from proximal blocks (Fig 3). The dispersal pattern observed in Arequipa is the likely result of relatively rare immigration to blocks without an established *T. cruzi* population and a low probability of establishing a population after immigration. These patterns support a model of gene flow in which city blocks are relatively homogenous, high-quality patches that allow within-block gene flow, separated by inhospitable barriers (*i.e.* roads and streets) that greatly reduce gene flow between adjacent blocks but that can act as conduits for inter-district gene flow through the aided movement of infected hosts or vectors.

The population genetic and phylogenetic analyses suggest that limited barriers to *T. cruzi* gene flow exist within city blocks. After *T. cruzi* establishes in a new block, the population readily disperses to neighboring houses within the block, likely within infected vectors that readily move between houses through shared walls [6,18]. *T. cruzi* population expansion within blocks is likely the result of the abundance of resources for both the parasite and its vector. For example, many households raise guinea pigs–which are particularly important reservoirs of *T. cruzi* [5,18–24] –in addition to housing dogs and humans, which offer an abundance of blood meals for the vector and competent hosts for the parasite [5]. These data suggest that blocks remain free of *T. cruzi* due to either low rates of immigration or low probabilities of establishment as populations thrive once established.

The discontiguous distribution of *T. cruzi* across city blocks suggests that dispersal between adjoining blocks is rare. *T. cruzi* was not detected in a majority of city blocks surveyed between 2008–2012 [5], despite many containing an active vector population (S5 Fig). Further, many blocks with a large *T. cruzi* population support only a single lineage suggesting most blocks were colonized only once (Fig 3). These data suggest either strong barriers to dispersal among city blocks or barriers that hinder the establishment of new populations. Prior investigations demonstrated that streets are a barrier to *T. infestans* dispersal, the primary vector of *T. cruzi* in the area [6,25], which is consistent with the behavior of other species in urban areas [26–28]. Nevertheless, dispersal of both *T. cruzi* and *T. infestans* has occurred between neighboring blocks, especially in older areas of the city. For example, a single *T. cruzi* lineage colonized neighboring Blocks 1–4 and a sample collected in Block 6 appears to have migrated from Block 7 (Fig 3). By contrast, the recently established population in Block 5 [5] has not invaded neighboring blocks (Fig 3). These data suggest that migration to adjacent blocks is relatively rare but can occur given sufficient time, supporting the hypothesis of a low inter-block dispersal rate.

The distribution of *T. cruzi* across city blocks is comparatively more fragmented than the distribution of its vector, suggesting that the parasite faces barriers to dispersal in addition to those encountered by the vector. These barriers could result from both life history characteristics of *T. cruzi*–such as inefficient transmission from vectors to hosts [20,29] –and specific

environmental features within blocks–such as the absence of vectors, competitive interactions with previously established *T. cruzi* lineages, or human-specific activity patterns–any of which could reduce the probability of establishing a population after introduction. The low transmission probability from vectors to hosts likely has a large impact on the probability of establishing a *T. cruzi* population because the parasite is not vertically [30] nor horizontally [23] transmitted between vectors. Therefore, *T. cruzi* may be unable to establish a population in a new block even if its infected vector successfully establishes a population. Indeed, many blocks contain an active vector population but do not harbor any *T. cruzi* (S5 Fig). While the dispersal of *T. infestans* is hindered by streets, there is some local between-block dispersal such that blocks within 5km of one another have an increased probability of harboring closely-related *T. infestans* [6,7]. Conversely, there is minimal evidence of local *T. cruzi* dispersal such that migration primarily occurs within blocks and between districts.

The geographic distribution of *T. cruzi* may also be impacted by the historical or current diversity of environmental features among city blocks, which could result in different immigration rates or establishment probabilities on each block [31]. For example, one block within the heavily sampled Mariano Melgar district (Fig 3, Block 6) contains six unique *T. cruzi* lineages (Fig 3), including one that immigrated from a proximal block (Block 7) and five that immigrated from different districts in the Arequipa region. By contrast, all other blocks have received no more than two independent introductions. The immigration and establishment of multiple lineages from adjacent and distant areas into one block may have resulted from unique environmental features or human activities such as the breeding or trading of guinea pigs [11].

The source of the immigrants that establish in each block do not appear to be more likely to originate from proximal blocks (four events) than from other districts (ten events) (Fig 3). That is, the closest relatives of the lineages inhabiting each block are often found across Arequipa and not necessarily the closest block with an active infestation. *T. cruzi* gene flow among districts is unlikely to be mediated by natural dispersal of infected vectors because *T. infestans* can only fly ~1km and do so only under stressful conditions such as starvation; an unlikely occurrence in human- and animal-occupied houses [32,33]. Interdistrict *T. cruzi* gene flow may instead be facilitated by the human-mediated transport of infected vectors or hosts. That is, human movement along the transportation infrastructure of the connected urban centers could facilitate the migration of *T. infestans* and reservoir species of *T. cruzi* among districts [34]. One potential hypothesis is human-mediated transport of guinea pigs, which are regularly brought to the study area from farms or purchased at a market (~8% of all individual guinea pigs). Given the small sample size, *T. cruzi* introductions cannot be statistically correlated with the number of guinea pigs imported onto infected blocks. However, the high rate of inter-district guinea pig importation is likely to increase the probability of inter-district *T. cruzi* dispersal given that a single infected guinea pig can infect many vectors over its years-long lifetime [5,21,24]. The hypothesis that guinea pigs can infect many vectors is further supported because the number of guinea pigs is correlated with the density of *T. infestans* (S4 Fig), showing that the presence of guinea pigs helps support a large vector population which increases the number of potential *T. cruzi* vectors.

The *T. cruzi* genotypes sampled in Arequipa are exceptionally uniform, as discussed previously [3] and as shown in previous reports of the broader region surrounding Arequipa [35]. Most *T. cruzi* sampled here were isolated from the only vector species in Arequipa, *T. infestans*, which reduces dataset biases for specific *T. cruzi* genotypes as *T. cruzi* from all infected mammal hosts must be transmitted to *T. infestans* to complete its life cycle. The *T. cruzi* samples were isolated from dogs and guinea pigs support this hypothesis as these samples are closely related to those isolated from vectors. Further, because *T. cruzi* must pass through *T. infestans*

to complete its life cycle, the parasite must infect a vector in order to proliferate in a new environment. Previous studies in domestic settings have found that the vector population harbors a more diverse *T. cruzi* population than hosts, suggesting that vectors carry the strains in local hosts in addition to strains not yet transmitted from vector to host [36]. One potential caveat is that *T. cruzi* genotypes residing long-term in mammalian hosts that have not transmitted to bugs will remained unsampled. Indeed, the only *T. cruzi* isolated from a human in this dataset has a unique genotype from those sequenced from bugs, dogs, and guinea pigs [3]. However, epidemiological models suggest that infected hosts are likely to transmit *T. cruzi* to vectors given sufficient time and vector abundance. That is, one infected human infects 21 vectors on average while 100 vectors on average are needed to infect one human [37,38]. By sampling *T. cruzi* primarily from vectors we likely capture the least biased representation of the diversity in the city in order to determine how a single population of *T. cruzi* migrated through Arequipa.

The data presented here demonstrate that urban areas such as Arequipa represent a patchwork of disconnected habitable areas, city blocks, with limited *T. cruzi* gene-flow among blocks and few barriers to gene-flow within blocks. The among-block gene flow that has occurred does not appear to be correlated with geographic distance, with many migration events occurring between districts that are unlikely to be traversed by vectors or mammalian hosts in the absence of human aid. These data support the hypothesis that considerable *T. cruzi* gene flow is facilitated by human transport of infected vectors or hosts. Such long-distance dispersal across the urban ecosystem may be possible for any species capable of exploiting human activity as a mechanism for migration. Without human-mediated migration, a *T. cruzi* population may be confined to individual blocks with infrequent dispersal across the barriers between blocks. These results highlight a difficulty in controlling dispersal of parasites like *T. cruzi* in urban areas. Long-term, city-wide vector control campaigns and continuous vector surveillance activities may be required to prevent spread of *T. cruzi* to new areas.

## Methods

### Ethics statement

The Institutional Animal Care and Use Committee (IACUC) of Universidad Peruana Cayetano Heredia reviewed and approved the animal-handling protocol used for this study (identification number 59605). The Universidad Peruana Cayetano Heredia Ethics Committee provided approval for human subject research and for the collection of bugs from households (identification number 52186). Human subject research participants provided written consent for the collection of bugs from his or her household and for human subject research. The IACUC of Universidad Peruana Cayetano Heredia is registered in the National Institutes of Health at the United States of America with PHS Approved Animal Welfare Assurance Number A5146-01 and adheres to the Animal Welfare Act of 1990 [3,39].

### Sample collection and study site

Briefly, 123 *T. cruzi* samples were isolated from infected *Triatoma infestans* bugs (N = 114), dogs (N = 2), and guinea pigs (N = 7) from houses throughout Arequipa, Peru (Fig 1A). Nearly half of the samples isolated from *T. infestans* (N = 56) were collected during intensive inspections of houses and surrounding areas along an established transect within the Mariano Melgar district (Fig 1B) in 2010 and 2011 as previously described [5]. An effort was made to inspect every house along this transect and collect bugs whenever possible such that *T. cruzi* isolates from most houses that harbored infected bugs were sequenced (S5 Fig). In 2011, in-depth interviews were conducted in each participating household that focused on animal

husbandry [11]. *T. cruzi* DNA was extracted from each sample and sequenced to an average depth of >50X as described previously [9].

## Whole genome assembly

Genomes were assembled using bowtie2 [40] and the most closely related reference genome, TcJR clone 4, obtained from TriTrypDB [41]. Only the 333 contigs longer than 10kb were used for the assembly to avoid spurious alignments for a total genome assembly that includes 28Mbp. This assembly largely excluded the extensive repeat regions found throughout the *T. cruzi* genome. Duplicate reads were removed using Picard MarkDuplicates [42].

## SNP calling

Individual genomic Variant Call Format (gVCF) files containing SNP data for each sample were generated using GATK HaplotypeCaller [42,43] following GATK's Best Practices procedure [44,45]. A joint genotype file containing all polymorphic sites from all samples was created using GATK GenotypeGVCF. Insertion-deletion mutations were not excluded in the joint genotype file. Polymorphic loci were hard-filtered by quality using GATK VariantFiltration, requiring Fisher strand bias (FS) <40, mapping quality >30, and quality by depth >10. Only loci for which all samples achieved a minimum depth of 20X and a Genotype Quality score greater than 40 were included. These filters maximized the number of polymorphic sites identified while ensuring that duplicate and triplicate sequences resulted in identical SNP datasets. The final consensus SNP panel included 9271 polymorphic sites.

## Population genetic analyses

VCF files were converted to plink format using vcftools [46] and plink [47]. Violin plots were used to visualize pairwise SNP distances between samples collected at different spatial scales using ggplot2 [48] in R [49]. AMOVA was performed using the *poppr* package in R [50]. Principle component analyses (PCA) were performed using Tassel 5 [51]. Phylogenetic relationships among *T. cruzi* isolates were inferred using a maximum clade credibility (MCC) phylogenetic tree reconstructed using BEAST 1.8.4 [52]. The best-fitting substitution model was determined using BEAST Model Test implemented in BEAST2 [53]. Starting from a UPGMA starting tree, the model implemented a Kimura 3-parameter substitution model with equal base frequencies, an Extended Bayesian Skyline coalescent tree prior [54], and a strict molecular clock. Log files were examined for convergence using Tracer v1.6.0 [55] ensuring ESS values above 200. The MCC tree was calculated using TreeAnnotator 1.8.4 with 10% burn-in and visualized in FigTree (available at http://beast.bio.ed.ac.uk). Phylogenetic analyses were performed using the BEAGLE library to increase computation speed [56,57]. Up to 100 independent iterations of ADMIXTURE [58] were run for each number of genetic clusters (K, ranging from 2 to 10) assuming linkage disequilibrium until the log-likelihood increased by less than $\varepsilon = 10^{-4}$ between iterations. The optimal number of clusters was estimated to be 4 by the cross-validation score averaged across 100 iterations (S6 Fig). The optimal alignment of the 100 iterations was calculated using CLUMPP [59].

## Animal husbandry and host dispersal data

Households across a transect in the Mariano Melgar district (N = 383) were surveyed for information including the number of guinea pigs owned, the number obtained from markets and farms, and the location of the markets or farms [11]. The number of guinea pigs imported from markets or farms from outside of Mariano Melgar was aggregated at the block level. *T.*

*cruzi* genomes and guinea pig data were available for 8 blocks in Mariano Melgar (Blocks 1–8). The minimum number of *T. cruzi* introductions to each block was estimated based on phylogenetic relationship (Fig 3) and was correlated to the number of guinea pigs imported. Blocks 1–4 were treated as one block because they share a single, recent common ancestor, indicative of a single introduction.

## Supporting information

**S1 Fig. ADMIXTURE analysis for K = 4 genetic clusters shows that similar genotypes tend to cluster within a block.** The colors of each pie chart represent the likelihood that a sample belongs to each of four genetic clusters. Each color (blue, yellow, orange, grey) represent a unique genetic cluster. There is significant clustering of identical genotypes within blocks 1–4, 5, and 7. Block 6 contains an exceptionally diverse population of *T. cruzi*. Block 8–12 have few samples per block, but the samples collected in each block are distinct from those collected elsewhere in the district. Blocks are displayed in a grid to maintain privacy.
(TIF)

**S2 Fig. Principal component analysis shows genetic similarity of T. cruzi collected from the same city block within the Mariano Melgar district.** PCA was calculated using 56 samples. The centroids of all samples from each city block with standard error bars are shown here. Each of the seven samples collected in Block 6 are represented by a red X because they occupy disparate portions of the PCA space. One sample collected in Block 7 is unique and is thus represented by its own point.
(TIF)

**S3 Fig. Using only presence/absence for each of 474 recombination events in place of genotype data, the *T. cruzi* isolates collected from the same block are genetically highly similar.** Violin plots show the number of pairwise differences in unique recombination events. Difference between all pairs of 123 samples, among pairs of samples collected in different blocks from the same district, and among pairs of samples collected in the same block, are shown. These results recapitulate the results found using genome-wide SNP data: The distribution of differences among samples between blocks among districts is larger than the distribution among all samples, the differences among samples within blocks are significantly smaller than expected given the diversity in the dataset ($p < 0.001$). The results that genetically-similar samples cluster in blocks while the overall diversity is distributed among districts do not vary regardless of the data set analyzed (recombination events or genome-wide SNP data).
(TIF)

**S4 Fig. The density of guinea pigs per block is correlated with the number of *T. infestans* vectors in Mariano Melgar.** The average number of non-human hosts and *T. infestans* vectors was counted for each house along a 2km transect in Mariano Meglar. The distribution of guinea pigs along the transect is similar to the distribution of the vector. The number of dogs and chickens remains relatively constant across the transect and is not correlated with vector population densities. The y-axis depicts the average count of each species per house. Approximate location along the transect of numbered blocks from Fig 1 are shown.
(TIF)

**S5 Fig. Many blocks with active *T. infestans* populations remain uninfected with *T. cruzi*.** Blocks containing infected vectors are colored red, blocks where only uninfected vectors were found are colored orange, and blocks where no vectors were detected are colored white [60]. Blocks where an infected vector was found but the parasite genome was not sequenced here

are represented by red blocks without block numbers. Blocks above the blue line were meticulously surveyed for parasites at four timepoints. Blocks below the blue line were surveyed by the Ministry of Health. *T. cruzi*-infected vectors were found in three additional blocks in Mariano Melgar outside of this region.
(TIF)

**S6 Fig. Cross-validation scores for each genetic cluster (K) averaged across 100 iterations.** Standard error bars are shown for each value. K = 4 was determined to be the optimal number of genetic clusters.
(TIF)

**S1 Table. Using only presence/absence for each of 474 recombination events in place of genotype data, AMOVA performed across Arequipan *T. cruzi* isolates shows significant structure within blocks.**
(DOCX)

**S2 Table. Sample collection location, source, and year.**
(DOCX)

**S1 Appendix. Maximum clade credibility phylogenetic reconstruction.** Newick file allows enhanced visualization of the phylogeny (Fig 1) using software including FigTree (available at http://beast.bio.ed.ac.uk). By default, tips are labeled with sample and branches in major clades are colored by location. Collection locations (district and block number within district) and dates for each sample can be found in S2 Table.
(NWK)

## Acknowledgments

The authors would like to thank Philippe Lemey for his advice regarding the use of Bayesian phylogenetics. The authors would also like to acknowledge Stephanie Seifert and Jill Devine for their assistance in the laboratory. The authors gratefully acknowledge the members of the Universidad Peruana Cayetano Heredia and the University of Pennsylvania Zoonotic Disease Research Lab in Arequipa, Peru, for their contributions, especially Carlos Condori and Luis Zamudio. The authors also thank Danitza Pamo, Jose Ylla, Jose Qusipe, Paul Picardo and Gabriela Bustamante for their contribution during the isolation and maintenance of the *T. cruzi* strains. In addition, the authors wish to acknowledge the advisory support provided by the following institutions: Ministerio de Salud del Perú (MINSA), the Dirección General de Salud de las Personas (DGSP), the Estrategia Sanitaria Nacional de Prevención y Control de Enfermedades Metaxenicas y Otras Transmitidas por Vectores (ESNPCEMOTVS), the Dirección General de Salud Ambiental (DIGESA), the Gobierno Regional de Arequipa, the Gerencia Regional de Salud de Arequipa (GRSA), the PanAmerican Health Organization (PAHO/OPS) and the Canadian International Development Agency (CIDA).

## Author Contributions

**Conceptualization:** Alexander S. F. Berry, Dustin Brisson.

**Data curation:** Alexander S. F. Berry, Claudia Arevalo-Nieto.

**Formal analysis:** Alexander S. F. Berry.

**Funding acquisition:** Michael Z. Levy, Dustin Brisson.

**Investigation:** Alexander S. F. Berry, Renzo Salazar-Sánchez, Jenny Ancca-Juarez, Michael Z. Levy.

**Methodology:** Alexander S. F. Berry, Claudia Chipana-Ramos, Melina Vargas-Maquera, Dustin Brisson.

**Project administration:** Renzo Salazar-Sánchez, Ricardo Castillo-Neyra, César Náquira-Velarde, Michael Z. Levy, Dustin Brisson.

**Resources:** Michael Z. Levy, Dustin Brisson.

**Supervision:** Katty Borrini-Mayorí, Michael Z. Levy, Dustin Brisson.

**Visualization:** Alexander S. F. Berry.

**Writing – original draft:** Alexander S. F. Berry, Michael Z. Levy, Dustin Brisson.

**Writing – review & editing:** Alexander S. F. Berry, Renzo Salazar-Sánchez, Ricardo Castillo-Neyra, Katty Borrini-Mayorí, Claudia Arevalo-Nieto, Claudia Chipana-Ramos, Melina Vargas-Maquera, Jenny Ancca-Juarez, César Náquira-Velarde, Michael Z. Levy, Dustin Brisson.

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
