## [Decision Letter · Decision Letter 0]

16 Dec 2019

Dear Dr Berry:

Thank you very much for submitting your manuscript "Dispersal patterns of Trypanosoma cruzi in Arequipa, Peru" (#PNTD-D-19-01885) for review by PLOS Neglected Tropical Diseases. Your manuscript was fully evaluated at the editorial level and by independent peer reviewers. The reviewers appreciated the attention to an important problem, but raised some substantial concerns about the manuscript as it currently stands. As detailed below, these concerns are mainly referred to the sampling and analysis of the data. In addition to the reviewers' comments, we would like to stress that it is known that even within 'restricted' domiciliary settings the presence of distinct parasite genotypes circulating in different hosts (i.e. humans, dogs, triatomines) has been described, which may contribute to the overall genotypic variability. Therefore, it should be considered (at least discussed) that sampling mostly triatomines may have biased somehow the results. These issues must be addressed before we would be willing to consider a revised version of your study. We cannot, of course, promise publication at that time. 

We therefore ask you to modify the manuscript according to the review recommendations before we can consider your manuscript for acceptance. Your revisions should address the specific points made by each reviewer. 

When you are ready to resubmit, please be prepared to upload the following:

(1) A letter containing a detailed list of your responses to the review comments and a description of the changes you have made in the manuscript.

(2) Two versions of the manuscript: one with either highlights or tracked changes denoting where the text has been changed (uploaded as a "Revised Article with Changes Highlighted" file); the other a clean version (uploaded as the article file).

(3) If available, a striking still image (a new image if one is available or an existing one from within your manuscript). If your manuscript is accepted for publication, this image may be featured on our website. Images should ideally be high resolution, eye-catching, single panel images; where one is available, please use 'add file' at the time of resubmission and select 'striking image' as the file type. 

Please provide a short caption, including credits, uploaded as a separate "Other" file. If your image is from someone other than yourself, please ensure that the artist has read and agreed to the terms and conditions of the Creative Commons Attribution License at http://journals.plos.org/plosntds/s/content-license (NOTE: we cannot publish copyrighted images). 

(4) If applicable, we encourage you to add a list of accession numbers/ID numbers for genes and proteins mentioned in the text (these should be listed as a paragraph at the end of the manuscript). You can supply accession numbers for any database, so long as the database is publicly accessible and stable. Examples include LocusLink and SwissProt.

(5) To enhance the reproducibility of your results, we recommend that you deposit your laboratory protocols in protocols.io, where a protocol can be assigned its own identifier (DOI) such that it can be cited independently in the future. For instructions see http://journals.plos.org/plosntds/s/submission-guidelines#loc-methods

While revising your submission, please upload your figure files to the Preflight Analysis and Conversion Engine (PACE) digital diagnostic tool, https://pacev2.apexcovantage.com/ PACE helps ensure that figures meet PLOS requirements. To use PACE, you must first register as a user. Then, login and navigate to the UPLOAD tab, where you will find detailed instructions on how to use the tool. If you encounter any issues or have any questions when using PACE, please email us at figures@plos.org.

We hope to receive your revised manuscript by Feb 14 2020 11:59PM. If you anticipate any delay in its return, we ask that you let us know the expected resubmission date by replying to this email.

To submit a revision, go to https://www.editorialmanager.com/pntd/ and log in as an Author. You will see a menu item call Submission Needing Revision. You will find your submission record there. 

Sincerely,

Carlos A. Buscaglia, PhD

Associate Editor

Eric Dumonteil

Deputy Editor

Reviewer's Responses to Questions

**Key Review Criteria Required for Acceptance?**

**Methods**

-Are the objectives of the study clearly articulated with a clear testable hypothesis stated?

-Is the study design appropriate to address the stated objectives?

-Is the population clearly described and appropriate for the hypothesis being tested?

-Is the sample size sufficient to ensure adequate power to address the hypothesis being tested?

-Were correct statistical analysis used to support conclusions?

-Are there concerns about ethical or regulatory requirements being met?

Reviewer #1: yes to all questions

Reviewer #2: My main concern refers to the origin of the samples which is not clear to me. Please indicate how many bugs from how many houses originated the 123 samples employed. I would suggest including in Suppl Table2 the host from which each sample was isolated, and the ecotope in the house where the bugs were collected from. Also, I would suggest ordering the information of this table by district, block and house (and not by district only as it is now) (i.e.: District, Block number, house number, host, number of samples).

**Results**

-Does the analysis presented match the analysis plan?

-Are the results clearly and completely presented?

-Are the figures (Tables, Images) of sufficient quality for clarity?

Reviewer #1: yes to all questions

Reviewer #2: Albeit I'm not a genetist, or perhaps because of that, I have some doubts regarding the genetic distances informed: In Results section (First paragraph). How can the average pairwise genetic distance between all samples be smaller (1726 SNPs) than the average pairwise distance between samples from different blocks within each district (1925 SNPs)?

In Results section (First paragraph of page 9) Can the limited genetic diversity within each block be due to a small number of houses sampled in each block? 

In Results section (First paragraph of page 10) I wonder if the observed pattern could be due to the fact that one or few houses have been sampled rather than “a block” effect. It is not clear to me if the effect of different houses within a block can be assessed with this sampling design. 

Regarding Figure 3’s legend. Is it not expected that districts that are more deeply (i.e Mariano Melgar and La Joya) sample exhibit a higher genetic diversity? 

Results section after the tables, page 12. Authors state “T. cruzi collected from different blocks tend to be unrelated regardless of spatial proximity (Fig 2)” Based on which analyses have the authors based this statement? I do not agree Fig 2 illustrates this statement. Also Supl. Fig 2 seems to show that the very distinct genotypes from Block 6 and Block 7 were very near.

Have you analyzed if there are any differences due to the year of sample collection? 

Discussion-First paragraph. “The genomic data support a pattern of migration in which dispersal proceeds with little impediment among houses within a city block soon after colonization of the block (Fig. 2 and S1).” How do the authors know that “soon after colonization”? I believe the authors have not done any temporal analyses. 

Discussion-Page 15 Second paragraph. Could the observed pattern be explained just by a recent colonization of the block (not by real barriers) and with time and transmission occurrence other neighboring blocks be colonized as in blocks 1-4?

**Conclusions**

-Are the conclusions supported by the data presented?

-Are the limitations of analysis clearly described?

-Do the authors discuss how these data can be helpful to advance our understanding of the topic under study?

-Is public health relevance addressed?

Reviewer #1: yes to all questions

Reviewer #2: (No Response)

**Editorial and Data Presentation Modifications?**

Reviewer #1: see general comments

Reviewer #2: -Line numbers are missing.

-I was not able to distinguish the colors in Figure 3. Could making lines thicker improve this figure?

-Please include axes legends of Supl Fig 4.

-M&M Please spell out VCF (in ref to VCF files).

Discussion. Page 14. “Interestingly, geographic proximity to blocks with an active population increases the establishment probability of that block only slightly (Figs 3 and S1)” What do authors consider “Active population” ? Please clarify.

 Please revise Table 2 legend. It includes, in fact among district data.

**Summary and General Comments**

Reviewer #1: in this study, Brisson et al use high resolution genomic tools (multilocus SNP) to survey the dispersal patterns of Trypanosoma cruzi among different districts of Arequipa (Peru). The metholology is sound and the conclusions are convincing.

The paper may be accepted as it is. However, the following points might be considered: 

(i) it would have been informative to characterize the isolates of the present study by comparison with the general classification of T. cruzi genotypes (DTU TcI-VI). It can be guessed that all Arequipa isolates are TcI. However, the adding of a few reference strains would have been informative. The relative genetic uniformity of T. cruzi genotypes of the present study clashes with former studies in which the presence of highly distinct DTUs has been evidenced in Arequipa (Brenière et al., 1985, Isoenzymic studies and epidemiological data of Trypanosoma cruzi from Arequipa (Peru), Pacific side. Ann. Soc. belge méd. trop. 65 (suppl.1): 63-66).

(ii) Isn't it possible to have an idea about the age of the most common ancestor (MCA) of these lineages? For example, using similar methodologies, yhe origin of the Escherichia coli ST131 clone has been dated a little bit before 2000 ("measurably evolving pathogens") (Stoesser N, Sheppard AE, Pankhurst L, De Maio N, Moore CE, Sebra R, et al. (2016) Evolutionary History of the Global Emergence of the Escherichia coli Epidemic Clone ST131. mBio 7: eo2162-15.). Such dating would be quite relevant in the present study.

typo page 17 line 14: "hwoever".

Reviewer #2: In this manuscript the authors describe the genetic diversity of Trypanosoma cruzi, the etiological agent of Chagas disease by using in depth sequencing. This is an interesting manuscript describing the distribution of genotypes at different scales in Arequipa. The results are original and clearly within the scope of the journal. There are however some major points that should in my opinion be addressed by the authors before the manuscript can be accepted for publication.

Major comments. 

Can we really speak of dispersal of T. cruzi? Is it not a distribution pattern analyses? 

As no human-related isolate has been analyzed in this study, I think this should be clarified in the title, in the abstract and throughout the discussion of the manuscript.

PLOS authors have the option to publish the peer review history of their article (what does this mean?). If published, this will include your full peer review and any attached files.

Reviewer #1: No

Reviewer #2: No

---

## [Decision Letter · Decision Letter 1]

18 Feb 2020

Dear Dr, Alexander Berry

We are pleased to inform you that your manuscript 'Dispersal patterns of Trypanosoma cruzi in Arequipa, Peru' has been provisionally accepted for publication in PLOS Neglected Tropical Diseases.

Before your manuscript can be formally accepted you will need to complete some formatting changes, which you will receive in a follow up email. A member of our team will be in touch within two working days with a set of requests.

Best regards,

Carlos A. Buscaglia, PhD

Associate Editor

Eric Dumonteil

Deputy Editor

Reviewer's Responses to Questions

**Key Review Criteria Required for Acceptance?**

**Methods**

-Are the objectives of the study clearly articulated with a clear testable hypothesis stated?

-Is the study design appropriate to address the stated objectives?

-Is the population clearly described and appropriate for the hypothesis being tested?

-Is the sample size sufficient to ensure adequate power to address the hypothesis being tested?

-Were correct statistical analysis used to support conclusions?

-Are there concerns about ethical or regulatory requirements being met?

Reviewer #1: answer to all queries: yes

Reviewer #2: (No Response)

**Results**

-Does the analysis presented match the analysis plan?

-Are the results clearly and completely presented?

-Are the figures (Tables, Images) of sufficient quality for clarity?

Reviewer #1: answer to all queries: yes

Reviewer #2: (No Response)

**Conclusions**

-Are the conclusions supported by the data presented?

-Are the limitations of analysis clearly described?

-Do the authors discuss how these data can be helpful to advance our understanding of the topic under study?

-Is public health relevance addressed?

Reviewer #1: answer to all queries: yes

Reviewer #2: (No Response)

**Editorial and Data Presentation Modifications?**

Reviewer #1: accept

Reviewer #2: (No Response)

**Summary and General Comments**

Reviewer #1: the authors have addressed all my points. The paper is acceptable as it is.

Reviewer #2: All my previous comments and concerns have been addressed. I thank the authors for their kind explications. I consider this version of the manuscript is suitable for publication.

PLOS authors have the option to publish the peer review history of their article (what does this mean?). If published, this will include your full peer review and any attached files.

Reviewer #1: No

Reviewer #2: No

---

## [Editor Report · Acceptance letter]

4 Mar 2020

Dear Dr. Berry,

We are delighted to inform you that your manuscript, "Dispersal patterns of Trypanosoma cruzi in Arequipa, Peru," has been formally accepted for publication in PLOS Neglected Tropical Diseases.

Best regards,

Serap Aksoy

Editor-in-Chief

Shaden Kamhawi

Editor-in-Chief
